# Seasonal and Monthly Patterns, Weekly Variations, and the Holiday Effect of Outpatient Visits for Type 2 Diabetes Mellitus Patients in China

**DOI:** 10.3390/ijerph16152653

**Published:** 2019-07-25

**Authors:** Yanran Huang, Jiajing Li, Hongying Hao, Lizheng Xu, Stephen Nicholas, Jian Wang

**Affiliations:** 1Center for Health Economics Experiment and Public Policy, School of Public Health, Shandong University, No. 44 Wenhuaxi Road, Lixia, District, Jinan 250012, China; 2NHC Key Laboratory of Health Economics and Policy Research (Shandong University), Jinan 250012, China; 3School of Economics and School of Management, Tianjin Normal University, West Bin Shui Avenue, Tianjin 300074, China; 4Guangdong Institute for International Strategies, Guangdong University of Foreign Studies, 2 Baiyun North Avenue, Baiyun, Guangzhou 510420, China; 5Top Education Institute, 1 Central Avenue, Australian Technology Park, Eveleigh, Sydney, NSW 2015, Australia; 6Newcastle Business School, University of Newcastle, University Drive, Newcastle, NSW 2308, Australia; 7Dong Fureng Institute of Economic and Social Development, Wuhan University, No. 54 Dongsi Lishi Hutong, Dongcheng District, Beijing 100010, China; 8Center for Health Economics and Management at School of Economics and Management, Wuhan University, 299 Bayi Road, Wuchang District, Wuhan 430072, China

**Keywords:** type 2 diabetes mellitus, seasonality, holiday, outpatient visits

## Abstract

Objective: To explore the seasonal and monthly patterns, weekly variations, and the holiday effect of outpatient visits for type 2 diabetes mellitus patients, as well as the influence of gender, age, and insurance type on variations. Methods: Data were obtained from the Shandong medical insurance database, including all outpatients in 12 cities of Shandong province in China from 2015 to 2017. The seasonal index (*S_t_*) was calculated in terms of seasons, months, and weeks by the moving average method. Results: A total of 904,488 patients received outpatient services during the study period. The seasonal indices of outpatient visits by type 2 diabetes patients were higher in autumn (108.36%) and spring (102.67%), while lower in winter (89.92%) and summer (99.04%), exhibiting an obvious seasonality. Gender and age had no effect on seasonal patterns. The month impacted the seasons patterns: January to February were the lowest and December the highest months of outpatient visits, complicating the seasonal patterns. We also identified a weekly pattern of outpatient visits. In addition, the outpatient visits for type 2 diabetes mellitus patients was also strongly affected by the Spring Festival, Lantern Festival, and National Day holiday periods. The type of medical insurance had a significant impact on outpatient visits. Conclusions: The outpatient visits for type 2 diabetes mellitus patients displayed seasonal patterns that were contradictory to the variations in blood glucose fluctuations found in previous studies and was also strongly affected by the holiday effect. The type of medical insurance impacted the pattern of outpatient visits.

## 1. Introduction

Seasonal patterns of type 2 diabetes mellitus (T2DM) have been noted extensively. T2DM seasonal patterns were related to glucose control [1,2,3,4], temperature [5,6,7,8], Vitamin D [9,10,11,12], birth season [13,14,15,16], insulin sensitivity [17,18,19], and hypoglycemia [20,21]. Seasonal variation of type 2 diabetes exists within components of the metabolic syndrome, such as glycaemia, which is significantly higher in winter attributed to multiple causes including seasonal checkups, low temperature, and affective disorder [6]. Some T2DM studies indicate that the changes of Hemoglobin A1c (HbA1c) level in T2DM patients followed an obvious seasonal pattern, which was significantly higher in winter (December, January, and February) than that in summer (June, July, and August) [22,23]. In addition, high temperatures are associated with increased rates of medical consultation, particularly in diabetics with cardiovascular disease [5]. A case-crossover analysis of more than 4 million UK GP consultations with T2DM sufferers found that there were increased odds of seeking medical consultation associated with high temperatures, which presented an autumn peak each year [5]. The number of diabetic events per season varied significantly in Hungary, with the acute myocardial infarction in diabetics occurring most in spring and least in summer [24]. Another Hungarian study found that the seasonality of T2DM incidence followed a sinusoidal pattern, the peak month was March and the trough month was August no matter the patient’s gender [25]. It was unclear whether outpatient visits for T2DM diabetics is consistent with season patterns in glycaemia or other seasonal rhythms.

Previous studies mainly used the detection data of outpatient services in hospitals or medical insurance to study the patterns of visiting doctors [5,25]. However, patients with chronic diseases may not seek medical treatment when the disease first presents unless the disease is aggravated or complications occur [26,27]. Therefore, the number of medical consultations is not a simple measure of the condition of the disease, but a reflection of the prevalence and severity of the disease, and the accessibility as well as the utilization capacity of health services [26]. The periodical effect can be seen in the treatment data from many medical institutions [28]. Previous studies have mainly focused on weekly or annual cyclical changes, ignoring the impact of public holidays [29]. Some studies attribute this “holiday effect” to the decrease of hospital staff during holidays and the lack of some special treatments and examinations during holidays [30]. So far, the literature on the diabetic holiday effect is relatively limited, and the results are contradictory [22,31]. The periodicity of the number of visits and the holiday effect must be taken into account simultaneously, so that the medical insurance management and hospital managers can continuously and effectively supervise the outpatient treatment during holidays and nearby days [32].

The aim of this study is to explore the seasonal patterns of outpatient visits by Chinese T2DM patients from multiple perspectives (season, month, week, holiday, and weekend), and compare the seasonal patterns of outpatient visits by T2DM patients with the seasonal changes of glycaemia found in previous studies. Better understanding the seasonality of T2DM leads to providing outpatient services more effectively and improving patients’ treatments.

## 2. Materials and Methods

### 2.1. Data Source

The T2DM outpatient data were obtained from a database of patients’ medical insurance during 2015–2017 in Shandong province, China. Located in China’s developed east coast and the lower reaches of the Yellow River, Shandong province has a warm temperate monsoon climate, with four distinct seasons. The database included information on the diagnosis, gender, age, medical insurance types, and medical expenses of T2DM outpatients visiting hospitals in 12 Shandong cities. The type of medical insurance consists of Urban Resident Basic Medical Insurance (URBMI) and Urban Employee Basic Medical Insurance (UEBMI), forming the basic medical insurance systems after new cooperative medical insurance merging into URBMI from 2015 in Shandong province. Since China’s basic medical insurance schemes covered 95% of the population, subjects included in this study accounted for almost all the Shandong T2DM outpatients during the study period.

### 2.2. Study Subjects

Based on the main diagnosis and International Classification of Diseases (ICD-10: E11-E14), 904,488 T2DM patients were identified from the medical insurance management database, including non-insulin-dependent diabetes mellitus, diabetic complications, and diabetic comorbidities, which constituted a 3-year dynamic cohort. Information on all outpatient visits made by T2DM patients, regardless of the reason, during the 3-year period (2015–2017) was extracted, resulting in 8,417,914 consultations.

### 2.3. Methods

Data were divided into several subgroups according to gender and types of medical insurance, and age stratified into six groups. Based on the raw data, Chi-square tests were used to analyze the effects of gender, age, and medical insurance types on differences in proportion between seasons. The moving average trend elimination method, namely the seasonal index analysis method of time series data, was adopted to estimate the seasonal indexes of each season, month, and week. According to Chinese meteorological classifications, spring was defined from March to May; summer from June to August; autumn from September to November; and winter from December to February. Starting from January 1 in each year, every 7 days was defined as one “week,” with each year having 52 weeks [32]. In order to facilitate a uniform comparison of the years, 2016 was a leap year, and February 29 was included in week 9 and December 31 was included in the last week of the year. To eliminate any variation due to differing month length, the average daily outpatient visits was used to replace the total outpatient amount for statistical analysis, such as chi-square test and seasonal indices calculation. In addition, as the number of weekends and statutory holidays may not be consistent within two months, if the influence of these factors collectively referred to as “holidays” is not excluded, the seasonal change trend of outpatient services cannot be truly reflected. Therefore, in the time series analysis of monthly and seasonal data, the influence of holiday factors must be removed from the original series, which is called “seasonal adjustment.” The non-parametric, scaling factor, method of the holiday effect adjustment was adopted [33,34,35] and the seasonal index calculated after eliminating the holiday factor, which more accurately reflects the basic medical treatment trend of diabetic patients.

The extracted data were analyzed using SPSS 22.0 software (SPSS Inc., Chicago, IL, USA). *p* < 0.05 was considered as statistically significant.

### 2.4. Ethical Considerations

The protocol for the research project was approved by the Public Health Ethics Committee of Shandong University (20190612) and it conforms to the provisions of the Declaration of Helsinki.

## 3. Results

### 3.1. The Outpatient Visits of T2DM Patients in Each Season

As shown in Table 1, data were collected on 904,488 subjects, comprising 50.76% males and 49.24% females, with an average age of 61.98 ± 10.7. The number of outpatient visits of T2DM patients in Shandong’s 12 cities was 2,312,145 in 2015, 2,884,437 in 2016, and 3,221,332 in 2017, showing an average annual growth rate of 18.22%. From Table 1, the type of medical insurance, Urban Resident Basic Medical Insurance (URBMI) or Urban Employee Basic Medical Insurance (UEBMI), had an effect on seasonal differences, but gender or age displayed no seasonality.

### 3.2. The Seasonal Index of T2DM Outpatient Visits in Each Season and Month

The dynamic analysis of outpatient visits’ in each season and month are shown in Table 2. Generally, the utilization of outpatient services for type 2 diabetics was clearly seasonal. The autumn seasonal index was the highest (108.36%) and winter the lowest (89.92%), indicating that autumn was the peak period for visiting doctors and winter the low period for doctor visits. The seasonal index of each season and month was statistically significant (*p* < 0.001).

The frequency of outpatient visiting was the minimum in January and February, and then gradually increased to peak in December, with a trough in July and August. Notably, the seasonal index of June was significantly higher than July and August, while the seasonal index of October was significantly lower than September and November. In winter, the number of outpatient visits was the lowest, but fluctuated the most. The seasonal indexes of December (110.49%), January (89.24%), and February (91.41%) were significantly different. The lower seasonal index for January, February, and October coincided with China’s most important festivals, the Spring Festival (January–February), Lantern Festival (February), and National Day holidays (October).

As Table 1 shows, the URBMI and UEBMI season indices varied. From Table 2, spring and autumn were the peak seasons for UEBMI insured outpatients, while the number of visits peaked in summer and autumn among URBMI outpatients. Outpatients insured by UEBMI had a seasonal index significantly higher than 100% (*p* < 0.001) in March, April, November, and December, and significantly lower than 100% (*p* < 0.001) in January, February, July, and August with no significant change in other months. The seasonal pattern for outpatients insured by URBMI was quite different. The seasonal index remained significantly above 100% from September to December and significantly below 100% from January to August except April (*p* < 0.001). Higher volatility in URBMI-insured outpatients’ visits suggest that their outpatient service utilization was more irregular than outpatients insured by UEBMI.

### 3.3. The Seasonal Index of T2DM Outpatient Visits in Each Week

Figure 1 displays that seasonal index by week, revealing a clear holiday effect with the frequency of the Spring Festival (7-day holiday in week 6) and National Day (7-day holiday in week 40) the lowest during the year. The seasonal index was 61.10% during the Spring Festival and 74.39% during the National Day holiday. The “weekly” seasonal index followed the same trend as the “monthly” seasonal index. In week 9 and week 41, there were two small peaks with the seasonal index reaching 105.92% and 115.90%, indicating a “catch-up” after the holidays. In week 52, the seasonal index reached the highest peak.

The seasonality of outpatient visits was influenced by medical insurance types. In general, URBMI-insured outpatients deviated more severely from the 100% baseline than UEBMI-insured patients. From week 1 to 6, the number of visits for URBMI-insured patients (53.39% in week 6) was significantly lower than UEBMI-insured outpatients (65.71% in week 6). The seasonal factors in week 7–39 were almost the same for both medical insurance types. From week 40 to 52, the number of visits for URBMI-insured outpatient visits (82.39% in week 40) was significantly higher than UEBMI-insured patients (70.02% in week 40). It appears that URBMI-insured outpatients may be more susceptible to the Spring Festival effect, while UEBMI-insured outpatients were more susceptible to the National Day holiday effect.

### 3.4. T2DM Outpatient Visits at Weekends

Figure 2 compares the actual visits with the expected visits throughout the week. The proportion of daily outpatient visits should be 1/7 (14.29%) if outpatient visits are evenly distributed over the seven days of the week, but this was not the case. The weekend outpatient visits were 10.67% on Saturdays and 9.04% on Sundays, while Mondays saw a catch-up, with a 17.12% visiting percentage. After Monday, outpatient visits trended downward through Thursday before a second peak on Friday, and a sharp decline over the weekend. The weekly pattern was almost the same for outpatients of different gender, age groups, and medical insurance types.

## 4. Discussion

Rather than studying outpatient visits by T2DM patients from a single perspective as in many previous studies, our study revealed yearly seasonal patterns of outpatient visits by T2DM patients, but also monthly, holiday, and weekly outpatient visit effects, and compared the seasonal patterns of outpatient visits by T2DM patients with the seasonal changes of glycaemia found in previous studies. Generally, the seasonal pattern of outpatient visits by T2DM patients fluctuated, with a high point in autumn and low point in winter. However, the monthly patterns revealed that seasonality was more complicated, with lows in January–February and July–August. The monthly patterns of high and low outpatient doctor visits also varied by insurance type, with spring and autumn the peak seasons for UEBMI insured outpatients and summer and autumn the peak seasons for URBMI outpatients. Two national holidays, the Spring Festival and National Day, also manifested a significant decrease in the seasonal indices. Focusing on weekly fluctuations, the frequency of clinic visits on weekends was significantly lower than that on working days, with weekday peaks on Monday and Friday.

In terms of seasonal patterns, it should be pointed out that the low seasonal index in winter is intertwined with the holiday effect, especially the Spring Festival effect. Due to this relationship, there is no uniform way to deal with the Spring Festival effect in our econometric model, which means our results need careful interpretation. The treatment of the Spring Festival effect was accounted the same weight as other holidays. Based on the above data analysis, our findings for seasonal patterns is consistent with several previous studies [5] that found summer and autumn were the peak periods of T2DM patients seeking medical treatment, which was related to heat-related consultations for diabetics with cardiovascular comorbidities [5,24]. Although medical studies have revealed that cold temperatures could lead to elevating glycosylated hemoglobin levels and acute complications of diabetes [1], the seasonality of diabetic medical service utilization was complicated by the seasonality of labor intensity, dietary habits, psychological factors, and exercise intensity [36]. Our seasonal patterns of outpatient services utilization in T2DM patients was counter to that of blood glucose. Importantly, our study found that there was a difference between the seasonal patterns of patients with UEBMI and those with URBMI. Higher summer seasonal index of URBMI patients indicated that the blood glucose management of URBMI patients might be more irregular with suggestive, but not definitive evidence, that URBMI patients suffer from a higher risk of exacerbation in hot seasons or weather.

The seasonal index by the month of the season fluctuated due to climate change, mood effects, and susceptibility to influenza. December was the peak month for outpatient service by T2DM patients, which was consistent with the conclusions the seasonality of HbA1c [1,23]. August had the lowest seasonal index for the differences, reflecting diet and exercise and increased insulin sensitivity in summer [37], which coincided with the results in the Hungary study [25]. Interestingly, in the busy March and April agricultural seasons in Shandong province, the seasonal index of URBMI patients, who were mainly farmers, was significantly lower than that of UEBMI patients. Whether there is an association between the two remains to be evaluated by further studies. According to our data analysis results, the seasonal index kept steady around 100% in other months except for alternate months, but URBMI patients had bigger outpatient volume in the alternate seasons, indicating that the illness was not managed routinely and effectively. Previous studies [38] have shown that T2DM patients have elevated levels of hormones that “antagonize” insulin, leading to blood sugar disorders and even complications or fatal damage, due to changes in temperature, weakness, and susceptibility to the flu at the turn of the seasons. Management of diabetes at alternate seasons is a challenge for diabetics and the management of diabetes in alternate seasons is helpful to avoid the occurrence of complications and improve the quality of life.

There may be another reason for the highest number of outpatient visits in December. The current policy on insurance reimbursements recalculates the deductible line at the beginning of every year according to the funds’ annual performance. From the perspective of benefit maximization, participants tend to make full use of their insurance policies, consuming all the remaining reimbursement quotas by the end of the year, and reserve medicines to avoid the restrictions on the deductible line in the coming year, leading to medical consumption increasing significantly at the end of every year [39]. Bao suggested that the periodicity of hospital admissions was likely to be correlated with the division of local medical insurance years [40], and there was a large periodic fluctuation in the number of hospital admissions in the months connecting the two medical insurance years. This correlation points to the need to reform medical insurance policies to improve the fairness and efficiency of health resources.

Our study has important health implications for T2DM outpatients. Outpatient visits were subject to non-health seasonal patterns. There was the reluctance of T2DM patients to seek medical treatment during China’s traditional Spring and Lantern Festivals leading to the winter period low point in T2DM medical care. During the festivals, people may lose normal control of their diet and exercise, eating too much high-calorie fried food and meat, when reuniting with friends and family; irregularly taking their prescribed drugs; and failing to timely monitor the changes of their glycaemia [36,37]. This is consistent with our data analysis results, which suggests that the low seasonal index during the holiday period may be partly due to the above reasons. A Hungarian study identified a similar result to our study [25]. There is evidence that the occurrence and risk of cardiovascular complications in post-holiday diabetics are higher than that of other periods [41]. The holiday season is also associated with small transient increases in both HbA1c and cholesterol [42]. Poor blood glucose control during the winter vacation cannot be reversed, and its cumulative effect is likely to lead to the annual increases of glycosylated hemoglobin in T2DM patients [22].

There may also be supply side factors in T2DM seasonality, where doctors’ working schedules lead to holiday effects. Fewer consultations were scheduled during the National Day and Spring Festival might explain fewer outpatient clinics visits during the national holidays. The holiday effect suggests the necessity to strengthen preventive health care and health education for diabetic patients before holidays (especially in winter) in order to reduce the blood glucose fluctuation and prevent complications. The URBMI-insured patients need more attention during the Spring Festival, while the UEBMI-insured patients should be focused on during the National Day holiday.

Focusing on the weekly fluctuations, both peak intensity on Monday and Friday might be caused by restricted outpatients’ appointments on the weekends. A similar result was also found in other studies [24] that the weekly peak of acute myocardial infarction in diabetics occurred on Monday, and gradually decreased to the weekend, with the substantial reduction over the weekend. In spite of different cultures and environments, the weekly pattern in Figure 2 was surprisingly consistent across different studies.

Our findings indicate that the frequency of seeking outpatient medical services by T2DM patients exhibits obvious seasonality in holidays, seasons, months, and weeks and the underlying mechanisms need further investigation. Consistent with previous research, low presentations at clinics during holidays, certain seasons, and months mean patients going without proper treatment. As suggested in previous studies, this unhealthy medical treatment mode should attract the attention of policy makers and clinicians. Physicians should focus on monitoring a series of physiological and pathological changes of patients during the low seasonal index period, timely correct any drug abuse or unreasonable drug use in T2DM treatment, and adjust prescriptions according to the different seasons. Moreover, the seasonal fluctuation of outpatient service utilization by T2DM patients also suggests that policy makers need to realize reasonable allocation and utilization of health resources through effective guidance. For example, T2DM patients could be provided with special educational pamphlet reminders on holiday periods, which could also lead to improvements in glycemic control. More concerning is the differences in T2DM presentations at clinics due to the type of medical insurance. Further research is required to explore the insurance phenomena.

Our research has some limitations. There is no uniform way to deal with the Spring Festival effect now, and econometric models cannot measure the length of effective period and the weight of effect, which depend on subjective judgment. Future studies should consider constructing three sections for the unevenly distributed Spring Festival model and assigning higher weights to further eliminate the impact of the Spring Festival holiday. Since the date of the Spring Festival is not fixed like the National Day holiday period, the statutory holiday intervals of the Spring Festival 2017 was 2.18–2.24 (equivalent to the eighth week) in 2015, 2.7–2.13 (equivalent to the sixth week) in 2016, and 1.27–2.2 (equivalent to the fifth week) in 2017. When we calculated the seasonal index on a weekly basis, there were inevitable errors due to the shifting Spring Festival dates in the calculation of the seasonal index of the Spring Festival holiday week, causing an overestimate of the seasonal index for the Spring Festival holiday week. In fact, during the Spring Festival holiday, the number of outpatient visits decreased sharply, and the actual holiday effect was larger than the value we presented.

## 5. Conclusions

We found that the outpatient services utilization in T2DM patients has seasonal, monthly, holiday, and weekly patterns, with the seasonal pattern of medical consultation exhibiting a peak in autumn and valley in winter, which is contradictory to the seasonal pattern of blood glucose rises found in previous studies. This phenomenon should arouse clinical attention. The visiting behavior of T2DM patients was also strongly affected by China’s long national holidays, season, monthly, and weekly patterns of demand for clinical services. The supply of doctor services might have impacted the pattern of holiday and weekend T2DM treatments. Importantly, we revealed that different types of medical insurance impacted the demand for outpatient services, which requires further investigation.

## Figures and Tables

**Figure 1 ijerph-16-02653-f001:**
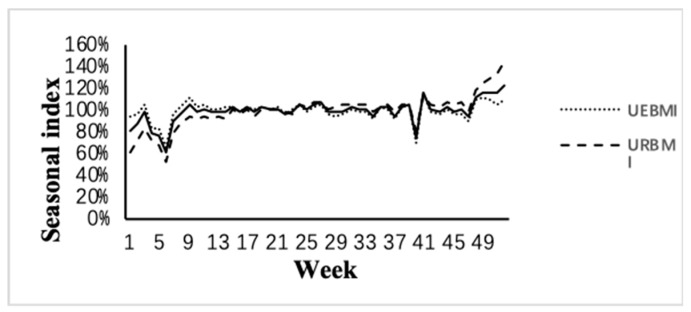
Seasonal index sequence.

**Figure 2 ijerph-16-02653-f002:**
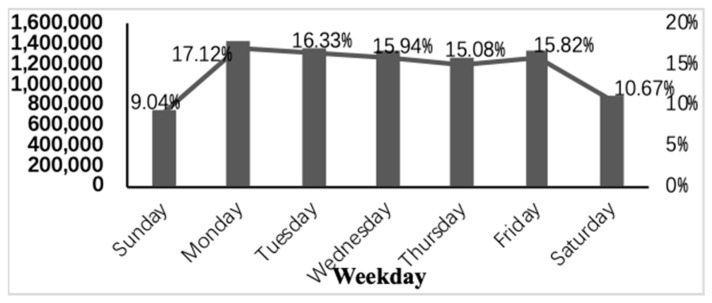
Weekend effect.

**Table 1 ijerph-16-02653-t001:** The frequency of outpatient visits in each season from 2015 to 2017.

Variables		*n* (%)					*p*-Value
		Spring	Summer	Autumn	Winter	Total	
		(*n* = 2,081,347)	(*n* = 2,208,152)	(*n* = 2,221,169)	(*n* = 1,907,246)	(*n* = 8,417,914)	
Gender							0.126
	Male	1,097,814	1,144,153	1,141,889	1,012,959	4,396,815	
		(24.97%)	(26.02%)	(25.97%)	(23.04%)		
	Female	983,533	1,063,999	1,079,280	894,287	4,021,099	
		(24.46%)	(26.46%)	(26.84%)	(22.24%)		
Age (years)							0.991
	≤39	42,587	47,578	50,353	41,769	182,287	
		(23.36%)	(26.10%)	(27.62%)	(22.91%)		
	40–49	184,676	198,453	204,523	178,179	765,831	
		(24.11%)	(25.91%)	(26.71%)	(23.27%)		
	50–59	552,585	590,805	595,650	515,629	2,254,669	
		(24.51%)	(26.20%)	(26.42%)	(22.87%)		
	60–69	773,037	824,329	829,764	700,552	3,127,682	
		(24.72%)	(26.36%)	(26.53%)	(22.40%)		
	70–79	435,002	451,967	449,163	387,464	1,723,596	
		(25.24%)	(26.22%)	(26.06%)	(22.48%)		
	≥80	93,460	95,020	91,716	83,653	363,849	
		(25.69%)	(26.12%)	(25.21%)	(22.99%)		
Medical Insurance Types							<0.001
	UEBMI	1,386,264	1,391,026	1,363,741	1,270,139	5,411,170	
		(25.62%)	(25.71%)	(25.20%)	(23.47%)		
	URBMI	695,083	817,126	857,428	637,107	3,006,744	
		(23.12%)	(27.18%)	(28.52%)	(21.19%)		

UEBMI: Urban Employee Basic Medical Insurance.

**Table 2 ijerph-16-02653-t002:** Seasonal indexes of outpatient visits in diabetics with Urban Employee Basic Medical Insurance (UEBMI) and Urban Resident Basic Medical Insurance (URBMI) (95% CI).

Season	UEBMI	URBMI	Total	Month	UEBMI	URBMI	Total
Spring	102.60% (100.45%, 103.55%)	102.93% (91.53%, 113.47%)	102.67% (99.05%, 105.71%)	March	105.36%	87.29%	98.98%
				April	109.08%	104.18%	107.43%
				May	99.66%	98.83%	99.49%
Summer	96.30% (94.45%, 97.55%)	103.65% (92.53%, 114.47%)	99.04% (95.44%, 102.09%)	June	99.86%	99.10%	99.71%
				July	93.31%	100.71%	95.24%
				August	92.66%	99.01%	94.85%
Autumn	104.00% (102.45%, 105.55%)	117.48% (106.03%, 127.97%)	108.36% (104.73%, 111.38%)	September	101.36%	104.43%	102.53%
				October	97.74%	105.69%	100.66%
				November	107.28%	114.63%	109.95%
Winter	97.10% (94.95%, 98.05%)	75.93% (65.03%, 86.97%)	89.92% (86.34%, 93.00%)	December	102.51%	124.73%	110.49%
				January	96.74%	75.47%	89.24%
				February	94.42%	85.93%	91.41%
Total	400%	400%	400%	Total	1200.00%	1200.00%	1200.00%

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
