# Peer review of "Seasonal and Monthly Patterns, Weekly Variations, and the Holiday Effect of Outpatient Visits for Type 2 Diabetes Mellitus Patients in China"

_ijerph, 2019, doi:10.3390/ijerph16152653_

Round 1

Reviewer 1 Report

This is an interesting paper that explores a relevant issue: patterns of use of health care services by the diabetic population. For this purpose authors explore and analyze the data using appropriate methods to assess the seasonal, monthly, holiday and weekly fluctuations.

The main problem with this paper is when authors try to address the second part of the objectives: "infer possible influencing factors" which is based on an assumption (that they do not test) that: "that the outpatient service utilization ... follows the same seasonal pattern with the changes of glycaemia".

Based on the date they incorporate in their models it is not possible to answer this specific objective. In fact, what authors try in the manuscript is to provide hypothesis on how several variables could explain their findings. 

However, although authors do not test the validity of these hypotheses is in this paper, they assume that their ideas are the explanation for the fluctuations identified in the visits of patients with diabetes, leading them even to make concrete recommendations about how to organize healthcare, clinical practice and what messages patients with diabetes  have to receive about their disease.

Author Response

Please find the attachment for "Cover Letter-International journal of evironmental research and public health"

Reviewer 2 Report

Comments for authors:

This study examines seasonality in diabetes-related outpatient visits in a province in China. A few comments/suggestions to help improve the manuscript:

1.       General: Please be consistent in referring to the study patients as “patients with type 2 diabetes mellitus” throughout the manuscript.

2.       Line 2-5, Title: “of type 2 diabetes mellitus outpatient visits”;

Line 21, Abstract: “of outpatient service utilization in type 2 diabetes mellitus patients”;

Line 34, Abstract: “utilization of outpatient services for patients with type 2 diabetes”

Comment: A clarification is needed to sort out the differences pointed out here. Is the study examining   a) outpatient visits where diabetes-related care was provided or b) diabetes-related outpatient visits by  patients with type 2 diabetes mellitus or c) “all” outpatient visits by patients with type 2 diabetes mellitus. Please clarify and be consistent throughout the manuscript.

3.       Line 24, Abstract: Please mention that seasonal indices (Si) were calculated for seasons, months, and weeks.

4.       Line 25, Abstract: “a total of 904,488 patients received outpatient services during the study period”.

Comment: How were study patients identified? From an insurance database or from data on outpatient visits itself? Was it a fixed and/or closed cohort (i.e. did you study fixed set of patients who had follow-up data for 3-year duration)? Was it a dynamic cohort?

5.       Line 26-28, Abstract: One suggestion would be to provide the seasonal indices for the four seasons in parentheses after each season and combine two redundant sentences.

6.        Line 30-33, Abstract: It appears that large deviations in seasonal indices (from 100%) in seasonal, monthly, and weekly patterns occurred only during holiday periods. Is it fair to attribute all variations to just “holiday affect”?  Or is the “holiday affect” an additional source of variation?

7.       Line 35, Abstract: “variations in blood glucose fluctuations”. Since, the blood glucose measures were not collected and analyzed as part of this study, one suggestion would be to say “previously published patterns of blood glucose fluctuations” to prevent any confusions.

8.       Line 37, Keywords: How come there is no mention of outpatient visits?

9.       Line 49 and Line 53, Introduction: References are needed.

10.   Line 53-55, Introduction: “T2DM seasonality followed a sinusoidal pattern” Can you please clarify what “T2DM seasonality” means. Is it glucose levels, clinical events, visits, hospitalizations?

11.   Line 67, Introduction: It appears Ref 29 doesn’t pertain to diabetes and was not retrievable.

12.   Line 73-74, Introduction: Refer to item 7.

13.    Line 79-85, Materials and Methods: A few additional lines on how the study cohort was identified and study setting should be added. Also refer to item 4. Was data on comorbidity, use of insulin/oral anti-glycemic medications, length of diabetes, and reason for visit available? Since, the readership is global, additional lines explaining the types of insurance would have provided better context. Were patients hospitalized included in the study? Was study limited to visits made to outpatient clinics attached to hospitals only?

14.   Line 85, Materials and Methods: “T2DM outpatients presenting for T2DM treatment during the study period”. Refer to item 2.  

15.   Line 87-102, Materials and Methods: Think “methods” and “statistical analysis” can be combined. Please arrange this section in the order in which the analyses were performed and presented in the manuscript.

16.   Line 94-96 and Line 100-102, Materials and Methods: It is not clear how many times statistical tests were performed and was it using raw data on visits or smoothed data.  Please resolve the differences.

17.   Line 109, Results: Parentheses is missing for S.D. of age. Assuming 10.7 represents S.D.

18.   Line 114-115, Results, Table 1: Use commas as the numbers are large and difficult to follow without commas. Suggest to replace p-value for insurance as < 0.0001 rather than leave it as 0.000.

19.   Line 121-137, Results: There is mentioning of significant differences of seasonal indices. Were these differences statistically tested?

20.   Line 139-155, Results: Peak SI was indicated to be in week 9 and week 41. But Figure 1 appears to show higher SI in weeks 48-52. Please clarify.

21.   Line 157, Results: It is not clear where and how the comparison of actual and expected visits were presented in Figure 2. Please expand.

22.   Line 166-266, Discussion and Conclusion: The seasonal and weekly patterns appear to be related to holidays and holidays-related migration; insurance types; insurance reimbursement policies; appointments scheduling. The seasonal indices for January/February vs December alone appear to be significantly different. Were differences for other months statistically significantly different? So, most of the differences appear to be attributable to holidays/scheduling/availability/insurance policies. Due to this linking the study findings to glucose levels and other factors, which were not collected and studied in the study appears to an overreach. Lastly, one would think we would find similar results for any chronic disease (ex. Hypertension, mental health illnesses). Would like to know what the authors have to say about this.

Author Response

(The authors gave the same response as above.)

Reviewer 3 Report

The study examined the seasonal and monthly patterns, weekly variations and the holiday effect of type 2 diabetes mellitus outpatient visits in China. However, from the described data, it can be assumed that seasonal variation is partly explained by the holiday effect. There are two main comments, Firstly, in analysis of seasonal effect, it needs to eliminate the effect of holiday effect. Secondly, to provide inferentially statistical support for the descriptive statistics, such as confidence interval or P value.

Author Response

(The authors gave the same response as above.)

Round 2

Reviewer 1 Report

The manuscript has improved and now may be published in its current form.

Author Response

Dear reviewer, 

thank you very much for your reviewing! We deeply appreciate your recognition of our research work.

Reviewer 2 Report

The authors have done a good job in responding to reviewers comments. I do not have any new comments to make.

Reviewer 3 Report

The authors answered most my concerns. But, before the discussion of seasonal or temperature effect, the holiday effectespecially the Spring Festivalshould be excluded or adjusted. The observed seasonal effect might contribute from the holiday effect. 

Author Response

Please find the attachment for document entitled "Cover Letter-International journal of evironmentalresearch and public health"
